# A Metabolic Enhancer Protects against Diet-Induced Obesity and Liver Steatosis and Corrects a Pro-Atherogenic Serum Profile in Mice

**DOI:** 10.3390/nu15102410

**Published:** 2023-05-22

**Authors:** Khrystyna Platko, Paul F. Lebeau, Joshua P. Nederveen, Jae Hyun Byun, Melissa E. MacDonald, Jacqueline M. Bourgeois, Mark A. Tarnopolsky, Richard C. Austin

**Affiliations:** 1Department of Medicine, Division of Nephrology, McMaster University, and the Research Institute of St. Joe’s Hamilton, Hamilton, ON L8N 4A6, Canada; platkok@gmail.com (K.P.); paulebeau91@gmail.com (P.F.L.); byunjh@mcmaster.ca (J.H.B.); macdome@mcmaster.ca (M.E.M.); 2Department of Pediatrics, Faculty of Health Sciences, McMaster University Medical Centre (MUMC), Hamilton, ON L8N 3Z5, Canada; nedervj@mcmaster.ca; 3Department of Pathology and Molecular Medicine, Faculty of Health Sciences, McMaster University Medical Centre (MUMC), Hamilton, ON L8N 5Z5, Canada; jmariebourg@icloud.com; 4Exerkine Corporation, MUMC, Hamilton, ON L8N 3Z5, Canada

**Keywords:** NASH, NAFLD, PCSK9, ROS ER stress

## Abstract

Objective: Metabolic Syndrome (MetS) affects hundreds of millions of individuals and constitutes a major cause of morbidity and mortality worldwide. Obesity is believed to be at the core of metabolic abnormalities associated with MetS, including dyslipidemia, insulin resistance, fatty liver disease and vascular dysfunction. Although previous studies demonstrate a diverse array of naturally occurring antioxidants that attenuate several manifestations of MetS, little is known about the (i) combined effect of these compounds on hepatic health and (ii) molecular mechanisms responsible for their effect. Methods: We explored the impact of a metabolic enhancer (ME), consisting of 7 naturally occurring antioxidants and mitochondrial enhancing agents, on diet-induced obesity, hepatic steatosis and atherogenic serum profile in mice. Results: Here we show that a diet-based ME supplementation and exercise have similar beneficial effects on adiposity and hepatic steatosis in mice. Mechanistically, ME reduced hepatic ER stress, fibrosis, apoptosis, and inflammation, thereby improving overall liver health. Furthermore, we demonstrated that ME improved HFD-induced pro-atherogenic serum profile in mice, similar to exercise. The protective effects of ME were reduced in proprotein convertase subtilisin/kexin 9 (PCSK9) knock out mice, suggesting that ME exerts it protective effect partly in a PCSK9-dependent manner. Conclusions: Our findings suggest that components of the ME have a positive, protective effect on obesity, hepatic steatosis and cardiovascular risk and that they show similar effects as exercise training.

## 1. Introduction

Metabolic Syndrome (MetS) consists of a cluster of interrelated risk factors, including obesity, hypertension, insulin resistance and dysglycemia, dyslipidemia and non-alcoholic fatty liver disease (NAFLD), and is among the leading causes of death worldwide [1,2,3]. Obesity is believed to be at the core of most metabolic abnormalities that fall under the MetS umbrella [2,4]. Approximately 75% of obese individuals are also afflicted by NAFLD, a condition often considered to be a hepatic manifestation of the MetS [2,5]. NAFLD is characterized by an accumulation of triglycerides in the liver that exceeds 5% of total liver weight and is comprised of non-alcoholic fatty liver (NAFL) and non-alcoholic steatohepatitis (NASH). While NAFL is essentially restricted to fat accumulation in the liver, NASH includes inflammation and evidence of hepatocyte injury. It is well established that NASH, characterized by hepatocyte injury/death, inflammation and fibrosis of the liver; can progress to cirrhosis, a terminal condition [5]. Limited evidence exists delineating the exact molecular mechanisms driving the progression of hepatic lipid accumulation; however, a multiple hit hypothesis suggests that oxidative stress, lipid peroxidation, Kupffer cell activation and adipocytokine alterations are central to this disease process [5]. Several studies also highlight the importance of endoplasmic reticulum (ER) stress in the development of NAFLD [6,7,8,9,10,11]. 

It is well-established that conditions associated with MetS drive systemic pro-oxidant and pro-inflammatory states, subsequently promoting hepatocyte lipid accumulation and injury [5,12,13,14]. An increase in the production of reactive oxygen species (ROS), as a result of cellular metabolic activities, can induce hepatocyte injury by damaging lipids, proteins and DNA in several cellular components including mitochondria. A persistent imbalance in the production of ROS and antioxidant molecules leads to enhanced lipid peroxidation, cytokine production and contributes to hepatocellular injury and fibrosis [15,16]. Furthermore, ROS production promotes the migration of pro-fibrogenic myofibroblast-like cells, originating from hepatic stellate cells, portal fibroblasts, or bone marrow derived cells, which is a critical process for the advancement of hepatic fibrosis observed in advanced NAFLD [17]. Reciprocally, excess free fatty acids trigger hepatic lipotoxic injury by facilitating the generation of several lipotoxic metabolites and molecules, including ROS [18]. Furthermore, altered redox homeostasis is sufficient to lead to ER stress, a major cellular stress pathway known to exacerbate the production of ROS [13]. 

In hepatocytes, pathological conditions that cause ER stress trigger the unfolded protein response (UPR) to regulate protein synthesis, glucose and lipid metabolism, as well as Ca^2+^ homeostasis [6]. The signaling cascades of the UPR comprise three master-regulator pathways, (i) the inositol-requiring enzyme 1 (IRE1)—X-box binding protein 1 (XBP1) arm, which is necessary for hepatic lipid metabolism through regulation of very low density lipoproteins (VLDL) secretion and lipogenesis [7,8,9]; (ii) the protein kinase RNA (PKR)-like ER kinase (PERK)—activating transcription factor (ATF) 4 arm regulates *de novo* lipogenesis via sterol regulatory element-binding protein (SREBP)-1 and fatty acid synthase (FAS) [10]; while (iii) ATF6 mediates de novo lipogenesis via SREBP-2 [11]. Protective at its core, UPR activation reduces secretory protein loading, enhances ER protein folding and increases clearance capacity of misfolded proteins from the ER. However, prolonged and/or severe ER stress contribute to NASH, inflammation and fibrosis, thereby leading to liver injury and dysfunction [6].

In the liver, proprotein convertase subtilisin/kexin 9 (PCSK9) levels impact the expression of genes involved in lipid metabolism, such as low-density lipoprotein receptor (LDLR), very low-density lipoprotein receptor, apolipoprotein B48, apolipoprotein A, and cluster of differentiation 36 (CD36) [19]. Recent studies have also highlighted the involvement of PCSK9 in the pathogenesis of NAFLD [20,21,22]. Clinical and pre-clinical evidence suggests a relationship between PCSK9 expression and hepatic steatosis [20,21,23]. Pursuant to these findings, PCSK9 knockout mice develop diet-induced hepatocyte lipid accumulation and injury [24]. Mounting evidence also demonstrates that there exists an association between ER stress and PCSK9 expression [25,26,27]. We have previously shown that ER stress can modulate the expression of PCSK9 in cultured hepatocytes [25] and that the loss-of-function PCSK9^Q152H^ variant is retained in the ER and protects against liver injury by increasing protein stability of the ER chaperones, including the glucose-regulated proteins of 78 kDa and 94 kDa (GRP78 and GRP94) [26,27]. Therefore, PCSK9 is thought to mitigate various mechanisms of hepatic steatosis and liver injury. 

Currently, there are no approved therapies for the treatment of NAFLD and the recommended management strategies include a combination of lifestyle modifications and weight management; however, physical activity guidelines are often met with poor adherence in obese populations [28,29], and sustained energy restriction is similarly difficult to maintain. Given the multi-factorial nature of the development of NAFLD/NASH, the specific mechanisms underpinning the beneficial impact of exercise on liver function are not fully elucidated [30]. Exercise training can lead to improvements in insulin resistance [31], reduction of SREBP-1c in circulating peripheral blood [32], improvements in fatty acid transport-, lipogenesis-, and β-oxidation-associated genes [30,33] and may reverse abnormal liver mitochondria [34]. The role of the PCSK9 pathway and exercise in the context of NAFLD has not yet been elucidated. The emerging pharmaceutical strategies for NASH are aimed at improving metabolic function, reducing steatosis, improving inflammation and halting or reversing fibrosis [28]. A growing body of evidence also illustrates a diverse array of naturally occurring antioxidants that attenuate MetS-associated NAFLD phenotype. Multiple cellular pathways involved in the development of NASH and fibrosis are regulated by a variety of naturally occurring metabolism-enhancing agents [35,36]. However, little is known about the (i) combination of these compounds on hepatic health and (ii) molecular mechanisms responsible for their effect. Here we explore the impact of a metabolic enhancer (ME), consisting of 7 naturally occurring mitochondrial enhancing agents, on diet-induced obesity, hepatic steatosis and atherogenic serum profile in mice. To this end we demonstrate that ME improved hepatic lipid homeostasis, attenuated obesity and improved atherogenic serum parameters in a PCSK9-dependent manner. Taken together, this 7-ingredient supplement designed to, in part, target mitochondrial function and facilitate weight loss, may function in a mechanistically similar manner to exercise. 

## 2. Materials and Methods

### 2.1. Animal Studies 

PCSK9^−/−^ mice were generated as previously described and were a generous gift from Dr. Nabil G. Seidah [37]; whereas, C57BL/6J mice were purchased from Jackson Laboratories (Stock No: 000664). All mice were housed in a 12 h light-dark cycle at ambient room temperature of 22 °C with free access to food and water. For studies examining the effect of diet-based ME on hepatocyte lipid accumulation and diet-induced obesity, 6-week-old male PCSK9^−/−^ and/or C57BL/6J mice were randomly placed on either a high fat diet (HFD; TD.06414, Envigo) containing 60% energy from fat or normal control diet (NCD; 2918, Envigo) (*n* = 6–10 per group). Following 6 weeks of HFD feeding, mice were placed on custom-formulation HFD containing 7 metabolic regulatory and mitochondrial enhancing agents, collectively termed ME, for an additional 4 weeks. ME components included seven well-established metabolic modulators as follows: green tea extract [38,39,40,41] (0.375% *w*/*w*), green coffee bean extract [42,43,44,45] (0.25%), alpha lipoic acid [46,47,48,49] (α-LA; 0.1%), forskolin [50,51,52] (0.005%), coenzyme Q10 [53,54,55,56] (CoQ10; 0.25%), beet root extract [57,58] (1%) and vitamin E [59,60,61,62,63] (0.20%). The diet was provided to the researchers at The Research Institute of St. Joseph Hospital (Hamilton, ON, Canada) by Exerkine Corporation (Hamilton, ON, Canada) in a blinded manner. Studies aimed at comparing the effect of exercise and ME were performed at the McMaster University Medical Centre (MUMC, Hamilton, ON, Canada). The group of mice exposed to a HFD and exercise served as a control group to compare the impact of conventional exercise with that of the ME on diet-induced obesity. Briefly, select groups of mice performed structured endurance exercise consisting of 5 min acclimation, 10 min warm up and 45 min exercise session 3 days a week (on alternate days) for the duration of the study. Following the completion of animal studies, de-identified samples were analyzed at The Research Institute of St. Joseph Hospital (Hamilton, ON, Canada). Samples were unblinded after analysis was completed. Findings of the experiments conducted at MUMC were independently validated in a cohort of PCSK9^−/−^ and C57BL/6J mice at The Research Institute of St. Joe’s Hamilton, using custom rodent diet formulation and experimental protocol identical to the studies carried out at MUMC. 

### 2.2. Hydroxyproline Assay

Fresh serum (100 μL) was mixed with an equal volume of 10 N concentrated NaOH and subsequently hydrolyzed for one hour. Samples were then neutralized with an equivalent volume of 10 N HCL and centrifuged for 5 min to remove undesired debris. Supernatants were mixed with reaction mixture and oxidation reagent mix as per manufacturer’s protocol (Abcam, Cambridge, UK; ab222941).

### 2.3. Immunohistochemical (IHC) Staining 

Paraffin blocks were cut into 4 μm thick sections, deparaffinized and dehydrated in 3 changes of xylene and 100% *v*/*v* ethanol, respectively. Following an endogenous peroxidase block, sections were then blocked in 5% *v*/*v* normal serum matching the species of secondary antibodies and incubated in primary antibodies for 18 h at 4 °C. Following incubation, the slides were exposed to biotin-labeled secondary antibodies (Vector Laboratories). Streptavidin-labeled horseradish peroxidase (HRP) solution (Vector Laboratories) and the developing solution (Vector Laboratories) were used to visualize the antibody staining. Slides were dehydrated and mounted in synthetic mounting medium (Electron Microscopy Sciences). Subsequently representative images were taken at 20× and 40× using a Nikon ECLIPSE Ci-L microscope (Nikon Instruments Inc., Melville, NY), equipped with a Nikon DS-Ri2 camera (Nikon Instruments Inc., Melville, NY, USA). All antibodies and working dilutions are listed in Appendix A.

### 2.4. Immunofluorescent (IF) and Histological Staining 

Tissues embedded in optimal cutting temperature (OCT) compound were cut into 10 μm cryosections, fixed for 1 h using 4% *w*/*v* paraformaldehyde, permeabilized for 15 min with 0.025% *v*/*v* Triton X-100 and subsequently blocked with 5% *w*/*v* bovine serum albumin (BSA). Slides were then incubated with the appropriate primary antibody overnight at 4 °C and stained with the appropriate Alexa Fluor-labeled secondary antibody (ThermoFisher Scientific, Waltham, MA, USA) and stained with DAPI (Sigma-Aldrich, St. Louis, MO, USA). For experiments aimed at examining hepatic lipid content, sections were prepared by the Core Laboratory at the McMaster University Medical Centre and stained with Oil-Red-O as described previously [64]. Slides were then mounted with Permafluor (ThermoFisher Scientific) and visualized using a light microscope (Nikon, Minato City, Tokyo) or a fluorescent microscope (EVOS-FL, ThermoFisher Scientific). All antibodies and working dilutions are listed in Appendix A. 

### 2.5. Immunoblots 

Immunoblot analysis was performed as previously described [65]. Briefly, tissue lysates were prepared using SDS lysis buffer containing a protease inhibitor cocktail (Roche, Basel, Switzerland). Protein concentrations were determined using a modified Lowry assay (Bio-Rad). Equivalent amounts of protein were resolved by SDS-PAGE and subsequently transferred to nitrocellulose membranes (Bio-Rad). The membranes were then blocked in 5% *w*/*v* skim milk, incubated with primary antibodies for 18 h at 4 °C, followed by HRP-conjugates secondary antibodies. Secondary antibodies were detected using enhanced chemiluminescence reagent (Amersham) and exposed using Konica Minolta X-ray film processor. All antibodies and working dilutions are listed in Appendix A. 

### 2.6. Quantitative Real-Time PCR

Quantitative real-time PCR was performed as described previously [24]. To summarize, total RNA was isolated using an RNA purification kit (ThermoFisher Scientific) according to the manufacturer’s instructions. A total of 2 μg of RNA was reverse transcribed using a High-Capacity cDNA Reverse Transcription Kit (ThermoFisher Scientific). PCR amplification was performed using fast SYBR Green (Applied Biosystems, Waltham, MA, USA). Relative transcript expression levels were calculated using the ΔΔCT method and normalized to 18S [66]. All primer sequences are listed in Appendix A. 

### 2.7. Quantification of Hepatic Triglyceride Content 

Hepatic triglyceride content was quantified as described previously [64]. Briefly, equal amounts of liver tissue were lysed in a mixture of hexane/2-propanol (Sigma-Aldrich) and incubated on an orbital shaker at 37 °C for 5 h. Lysates were then centrifuged for 5 min at 12,000 rpm to isolate the lipid-containing liquid fraction. Lipid content was quantified using a colorimetric triglyceride assay (Wako Diagnostics, Mountain View, CA, USA) according to the manufacturer’s instructions. 

### 2.8. Steatosis Score 

Steatosis grade was determined by a trained anatomical pathologist (JB) according to the guidelines summarized in Appendix A. The pathologist was blinded during the quantification process and the code was revealed after data analysis was complete. 

### 2.9. ELISAs and Alanine Aminotransaminase (ALT) Assay 

Circulating levels of PCSK9 (MPC900, R&D), ApoB (ab230932, Abcam) and ApoA1 (ab238260, Abcam) levels were measured using a commercially available mouse ELISA kits. Plasma ALT was measured using a commercially available colorimetric assay (ab241035, Abcam). All assays were performed according to the manufacturer’s instructions.

### 2.10. Statistical Analysis 

Data are presented as mean ± standard deviation and were analyzed using the unpaired Student’s t-test or one-way ANOVA with Tukey multiple comparison testing, where *p* < 0.05 considered significant. Details of biological replicates are listed in figure legends. The graphical abstract was created with BioRender.com (agreement number TS25DLOLAJ).

## 3. Results

### 3.1. ME Inhibits Diet-Induced Weight Gain and Hepatocyte Lipid Accumulation

Previous evidence demonstrated that individual components of the ME used in this study improve lipid metabolism and attenuate diet-induced obesity [35,36], and the specific supplement combination used in the current study confirmed the combined efficacy of the ME [67]. In line with these findings, we confirmed that dietary supplementation with ME attenuated HFD-induced weight gain (Figure 1A,B; * *p* < 0.05, *n* = 5–6). Additionally, the size and weight of inguinal, gonadal, and brown adipose tissue (IAT, GAT, BAT) were significantly smaller for the ME supplementation group (Figure 1C,D; * *p* < 0.05, *n* = 5–6). Consistent with gross morphological observations, ME resulted in smaller adipocyte size and lipid droplet content in GAT and BAT, respectively (Figure 1E). 

Given that the liver plays an important role in lipid metabolism and that hepatic steatosis is commonly associated with obesity [5,14], we next examined the livers of mice fed HFD+ME vs. HFD+Exercise group. Consistent with observation in adipose tissue, ME supplementation reduced hepatic weight and improved morphologic appearance of the liver compared to the HFD-fed mice (Figure 2A,B; * *p* < 0.05, *n* = 5–6). Strikingly, analysis of hepatocyte lipid (steatosis) content revealed that the addition of ME to the HFD reduced steatosis to a similar extent as observed in the HFD+Exercise group (Figure 2C–E; *p* < 0.05, *n* = 5–6). Parameters for determination of steatosis grade are summarized in Appendix A. In line with these observations, quantitative real-time PCR analysis also showed that ME and exercise both attenuated the expression of known drivers of hepatocyte lipid accumulation, including SREBP1, adipocyte differentiation-related protein (ADRP), hepatocyte nuclear factor 1 α (HNF1α) (Figure 2F; *p* < 0.05, *n* = 5). Overall, these observations demonstrated that in addition to its anti-obesogenic effect, ME also mitigated hepatic steatosis. 

### 3.2. ME Attenuates UPR Activation, Apoptosis and Liver Injury to a Similar Extent as Exercise

Activation of ER stress and dysregulation of the UPR are implicated in the pathogenesis of NAFLD and NASH [6,13]. Thus, we next examined the expression of protein markers associated with ER stress and apoptosis in mice fed NCD, HFD, HFD+ME, as well as mice fed HFD in combination with the exercise regiment. Consistent with the observed increase in hepatic lipid content, IF staining revealed an attenuation of ER stress markers GRP78 and GRP94 in the livers from HFD+ME and HFD+Exercise mice (Figure 3A). Additionally, real-time PCR analysis and immunoblotting demonstrated a decrease in the expression of markers of ER stress and apoptosis, GRP78, GRP94, spliced XBP1, IRE1α, ATF4, PERK, C/EBP homologous protein (CHOP) and cleaved caspase 1 [68] in the livers from HFD+ME and HFD+Exercise mice (Figure 3B,C; *p* < 0.05, *n* = 5). Similarly, ME attenuated the expression of GRP78 and GRP94 in GAT and BAT (Appendix A). 

Given that NASH can result in progressive hepatic fibrosis and end stage liver cirrhosis [5,14], we next examined the expression of markers of fibrosis and hepatic injury in the livers from HFD+ME and HFD+Exercise mice. IF staining revealed a reduction in the expression of fibronectin in the livers from these mice (Figure 4A). Hepatic fibrosis was also confirmed using picro-sirius red (PSR) and trichrome, which are conventional stains used for the study of hepatic fibrosis known to stain collagens. Consistent with fibronectin staining, an increase in PSR and trichrome staining was observed in mice treated with HFD, but not in mice treated with HFD+ME compared to the NCD control group (Figure 4B and Appendix A). A circulating marker of fibrosis (hydroxyproline) was also examined; treatment with ME led to a numeric reduction in hydroxyproline compared to the HFD group (Appendix A). In line with an attenuation of liver injury, ME and exercise were both able to reduce serum levels of ALT in mice fed HFD (Figure 4C; *p* < 0.05, *n* = 8–12). Additionally, apoptosis and inflammation were reduced in the livers from HFD+ME and HFD+Exercise mice (Figure 4D,E; *p* < 0.05, *n* = 5). Similar results were also observed using a TUNEL assay for apoptotic cells (Appendix A). Collectively, these data suggest that the hepatoprotective effect of ME contributes to the improved liver function and reduction in hepatic injury, an effect similar to that seen with exercise. 

### 3.3. ME Improves Pro-Atherogenic Serum Profile

Mounting evidence indicates that NAFLD is strongly associated with increased risk of cardiovascular disease [69,70]. For that reason, we next examined the serum profile of mice fed NCD, HFD, HFD+ME, as well as mice fed HFD in combination with exercise regimen. Both ME and exercise attenuated serum PCSK9 and ApoB (a circulating marker of LDL cholesterol), while ApoA1 levels remained unchanged (Figure 5A–C; *p* < 0.05, *n* = 6–12). Consistent with these observations, the mRNA expression of PCSK9 and SREBP2 were also downregulated in the livers from HFD+ME and HFD+Exercise mice (Figure 5D; *p* < 0.05, *n* = 5). In line with the reduction in the expression of PCSK9 and SREBP2, IF staining revealed maintenance of the expression of LDLR in the livers from HFD+ME and HFD+Exercise mice as compared to WT mice vs. the lower abundance in the HFD mice (Figure 5E). 

To further examine the correlation between PCSK9 and the metabolic profile observed in mice fed HFD, we also utilized a model of diet-induced obesity in PCSK9^−/−^ mice [24]. Similar to our previous observations in WT mice, ME significantly reduced weight gain and adiposity in PCSK9^−/−^ mice (Figure 6A,B; *p* < 0.05, *n* = 5). The ME also improved gross morphological appearance of hepatic tissue from PCSK9^−/−^ mice (Figure 6C,D). In line with these observations, IHC staining revealed that HFD+ME increased LDLR expression in the livers from C57BL/6J mice, while no changes were observed in the livers from PCSK9^−/−^ mice (Figure 6E,F; *p* < 0.05, *n* = 5–6). Similarly, IHC staining and quantitative real-time PCR analysis revealed that ME significantly reduced the expression of CD36 in livers from C57BL/6J in response to HFD, while this effect was attenuated in the livers from PCSK9^−/−^ mice (Figure 6E,F, Appendix A; *p* < 0.05, *n* = 5–6). Given that both LDLR and CD36 are downstream targets of PCSK9 [19], these findings suggest that ME exerts its protective effect partly in a PCSK9-dependent manner. 

## 4. Discussion

MetS comprises a cluster of interrelated diseases, characterized by abdominal obesity, dyslipidemia, insulin resistance, fatty liver disease and vascular dysfunction; and is a major public health concern with a prevalence of 20–45% [3] throughout all epidemiological studies. Collectively, cross-sectional and interventional studies demonstrate that each component of MetS is favourably influenced by interventions that include physical activity [71]. While exercise can both prevent and mitigate several components of MetS it remains underutilized by a large proportion of the general public [71,72]. For that reason, alternative and complementary interventional strategies are a potential solution that may bridge this gap. A growing body of evidence also illustrates that a diverse array of naturally occurring antioxidants, phytosterols and mitochondrial enhancing agents can improve a number of physiological, biochemical and metabolic parameters, thereby contributing to an attenuation of pathologies associated with MetS [35,36]; however, mechanistic studies to evaluate the specific modes of action and interactive effects of combining such agents is currently lacking. In line with previous observations, we confirmed that a diet-based ME decreased body weight and adiposity in mice [67]. Additionally, we demonstrated an attenuation of hepatocyte lipid accumulation in mice fed diet-based ME. In line with these observations, we show that the specific ME used herein reduced hepatic ER stress, fibrosis, apoptosis and inflammation, thereby improving overall liver health. In addition to its hepatoprotective effect, ME ameliorated the HFD-induced pro-atherogenic profile. Moreover, our data suggest that the ME exerts its protective effect on the atherogenic serum profile in a PCSK9-dependent manner. 

The accumulation of visceral adipose tissue, characteristic of MetS, drives systemic pro-oxidant and pro-inflammatory states [12,73]. Epidemiological, clinical and animal studies have reported the role of oxidative stress in the pathogenesis of obesity and associated diseases, such as NAFLD [12,13]. In addition to oxidative stress, ER stress is also linked to various features associated with NAFLD and can occur independently or as a result of increased ROS production [13,14]. Herein, we demonstrate that a dietary supplement, consisting of seven naturally occurring compounds (some with antioxidant properties) mitigates ER stress, inflammation, apoptosis and liver injury. These findings are consistent with the previously demonstrated ability of the components of the ME to reduce hepatocyte lipid accumulation by attenuating these pathological processes. Studies showed that (-)- epigallocatechin-3-gallate (EGCG), one of the major components of green tea extract, diminished HFD-induced hepatocyte lipid accumulation in mice [39,74]. Similarly, green coffee bean extract attenuated hepatocyte lipid accumulation and ER stress; an effect that was attributed to high chlorogenic acid found in unprocessed coffee beans [43,75]. Likewise, powerful endogenous antioxidants such as CoQ10 and α-Lipoic acid both inhibited hepatocyte lipid accumulation and liver injury in rodent studies [56,76]. Among the various compounds found in red beetroot extract, betalains are widely accepted to possess strong anti-oxidative, anti-apoptotic and anti-inflammatory properties [57]. Dietary forskolin has also been shown to attenuate oxidative stress, inflammation and hepatic injury [51]. Vitamin E possesses anti-oxidative, anti-obesogenic, anti-hyperglycemic and anti-inflammatory properties [77,78]. In a clinical setting, the use of vitamin E as a monotherapy or in combination with other agents resulted in improved liver biochemistry and histology in patients with NAFLD and NASH [79,80]. Overall, our data suggests that a combination of the aforementioned ME components has a significant protective effect against hepatic ER stress, apoptosis and inflammation. In adipose tissues, our research group recently demonstrated that treatment with the ME led to an upregulation of white and brown adipose tissue mRNA transcripts associated with mitochondrial biogenesis, browning, fatty acid transport, and fat metabolism. Increased mitochondrial oxidative phosphorylation protein expression and in vivo fat oxidation was also observed in white adipose depots [79]. Importantly, however, a net reduction in food consumption was also observed in the HFD+ME group compared to the HFD group. Appetite suppression therefore likely represents an additional mechanism by which the ME described in this study acts to reduce the overall lipotoxic phenotype observed in the ME+HFD group [79]. 

Hepatic lipid metabolism is regulated by various transcription factors and nuclear receptors that work in concert to maintain hepatic lipid homeostasis [81]. Recent evidence shows that perturbations in ER stress signaling cascades can lead to an increase in hepatic *de novo* lipogenesis and lipid droplet formation by upregulating the expression of key lipogenic transcription factors [82,83]. To this end, we demonstrate that in addition to an attenuation of ER stress and inflammation, the ME and exercise interventions each reduced the expression of lipid regulatory genes, such as SREBP1, ADRP, HIF1α. These effects likely occurred as a direct result of (i) attenuation of hepatic ER stress and injury, as well as the (ii) anti-adipogenic effect of several components of ME on fatty acid synthesis, lipogenesis, lipolysis and *β*-oxidation in metabolically active tissues. EGCG and chlorogenic acid were found to prevent hepatic steatosis by increasing lipid oxidation in the liver [75,84]. Although there currently exists limited information on the effect of forskolin on lipolysis in the liver, in vitro and in vivo studies demonstrate that forskolin stimulates lipolysis in adipose, thereby decreasing fat mass [52,85]. There also exists a well-established relationship between vitamin E and CD36 in the context of atherogenesis, whereby vitamin E prevents atherogenic lesion formation by inhibiting the expression of CD36 [63,86]. Consistent with these findings, we demonstrate that ME attenuated the expression of a well-established driver of hepatocyte lipid accumulation, CD36. Importantly, several studies have also characterized the protective role of ME components in metabolic tissues other than liver, which may in turn have an indirect positive impact on liver health [57,77,87]. 

Abundant evidence also indicates a strong association between NAFLD and increased risk of cardiovascular disease [69,70]. Herein, we report that in addition to anti-obesogenic and hepato-protective effects, ME and exercise improved HFD-induced pro-atherogenic serum profile in mice. In support of this notion, pre-clinical and clinical studies have demonstrated cardio-protective effects of several ME components [88,89]. Cui et al., showed that green tea, rich in EGCG, improved serum lipid profile and reduced circulating PCSK9 in human and rodent studies. Mechanistically, the authors concluded that EGCE reduced PCSK9 abundance and LDL-C levels by downregulating the expression of HIF1α and upregulating the expression of hepatic LDLR [88]. These findings are consistent with previous studies demonstrating (i) a decrease in plasma cholesterol in PCSK9^−/−^ mice [37] as well as (ii) a reduction in serum cholesterol in mice and non-human primates following the treatment with an anti-PCSK9 neutralizing antibody [90]. In line with this observation, we demonstrated that ME and exercise each independently improved serum lipid profile and attenuated PCSK9 levels in HFD-fed mice. Vitamin E is another well-established regulator of cholesterol metabolism thought to have a protective effect against CVD [61,63,86]. Studies in human subjects and rodent models of disease demonstrated that beetroot extract is also a powerful modulator of atherogenesis capable of improving serum lipid profile and vascular function [57]. Together, the data presented in this study suggest that in addition to its anti-obesogenic and anti-steatotic effect, ME significantly improves serum lipid profile and reduces levels of pro-atherogenic PCSK9, which likely attenuates the risk of cardiovascular disease.

## 5. Conclusions

It has long been established that exercise is an important countermeasure for many metabolic abnormalities associated with MetS [71]. Although there are theoretical concerns that antioxidants within ME can attenuate some of the health benefits of exercise by blunting the ROS signaling [91], the present study demonstrates that diet-based ME and exercise independently improve adiposity and hepatocyte lipid accumulation in mice. Mechanistically, ME and exercise reduced hepatic ER stress, fibrosis, apoptosis, and inflammation, thereby improving overall liver health. Because the benefits conferred by the ME are inclusive of other metabolic tissues (i.e., adipose tissue), and that such tissues are inter-dependent, the exact mechanism by which the ME mitigates hepatic steatosis remains to be determined. In addition to a reduction in diet-induced hepatic steatosis and adiposity, Our results also show that in addition to its hepato-protective effect, the ME improved a pro-atherogenic serum profile by lowering circulating PCSK9 and ApoB. Thus, our findings suggest that components of ME have a positive, and likely cumulative effect on obesity, hepatic steatosis and cardiovascular risk. Given our pre-clinical findings, future clinical studies are needed to establish whether obese patients may benefit from ME supplementation to improve their weight loss strategy. 

## Figures and Tables

**Figure 1 nutrients-15-02410-f001:**
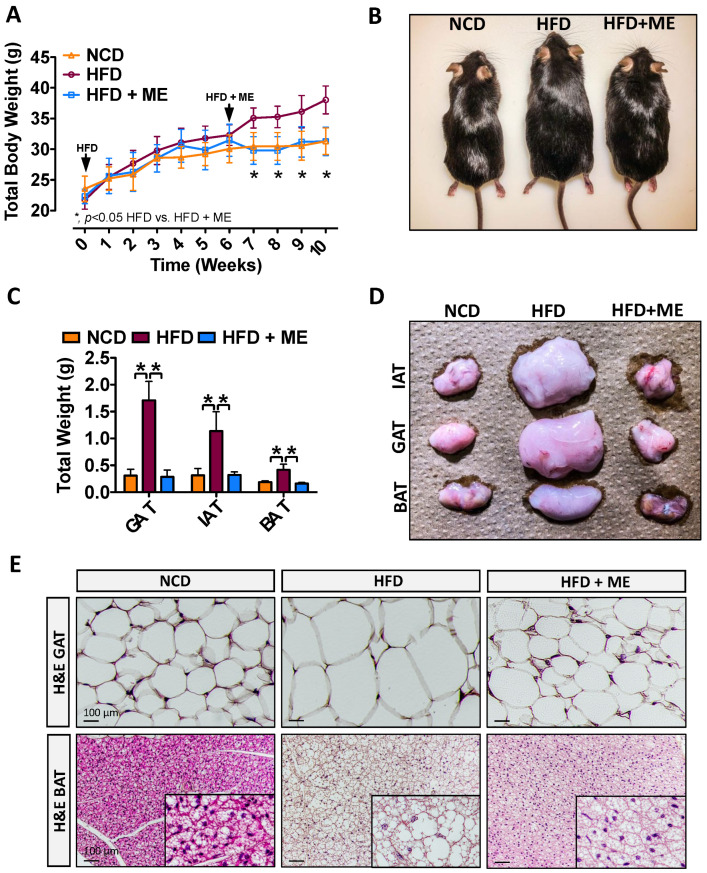
Metabolic enhancer protects against diet-induced weight gain and adiposity. (**A**) Mean body weight and (**B**) macroscopic appearance of wild-type mice fed either NCD, HFD, or HFD+ME (*n* = 5). (**C**) Mean weight and (**D**) macroscopic appearance of the representative IAT, GAT, and BAT (*n* = 5). (**E**) H&E staining of the GAT and BAT. Scale bars, 100 µm. All data are shown as mean ± SD. *, denotes *p* < 0.05 by one-way ANOVA with Tukey multiple comparison testing using Prism 6 (GraphPad, San Diego, CA, USA).

**Figure 2 nutrients-15-02410-f002:**
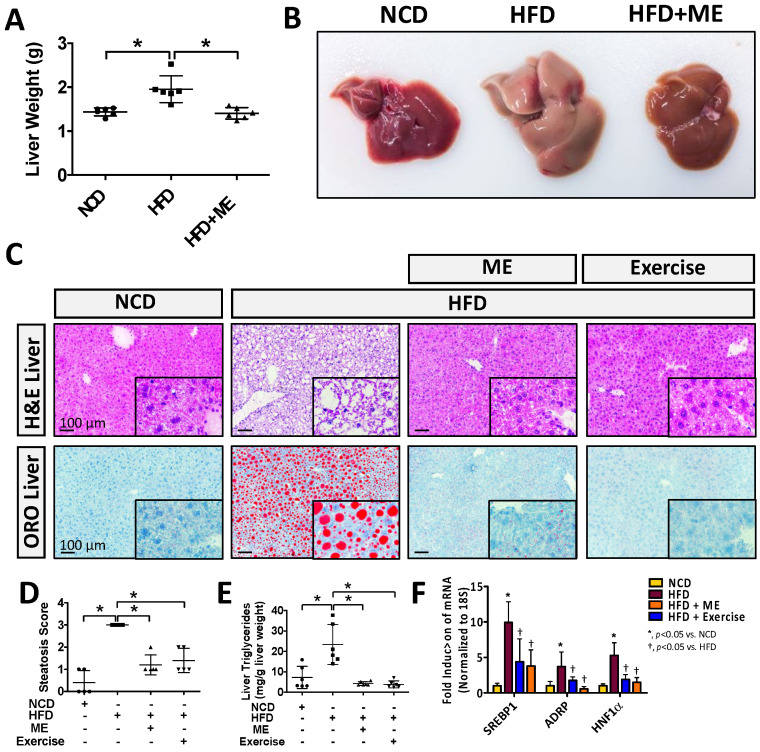
Metabolic enhancer and exercise both independently attenuate diet-induced hepatic steatosis. (**A**) Mean hepatic weight and (**B**) macroscopic appearance of livers from wild-type mice fed either NCD, HFD, or HFD+ME (*n* = 5). (**C**) H&E and ORO staining of livers. (**D**) Steatosis score (*n* = 5). (**E**) Hepatic triglyceride content (*n* = 6). (**F**) Quantitative real-time PCR analysis of hepatic mRNA abundance of indicated genes (*n* = 5–6). Scale bars, 100 µm. All data are shown as mean ± SD. *, denotes *p* < 0.05 by one-way ANOVA with Tukey multiple comparison testing using Prism 6 (GraphPad).

**Figure 3 nutrients-15-02410-f003:**
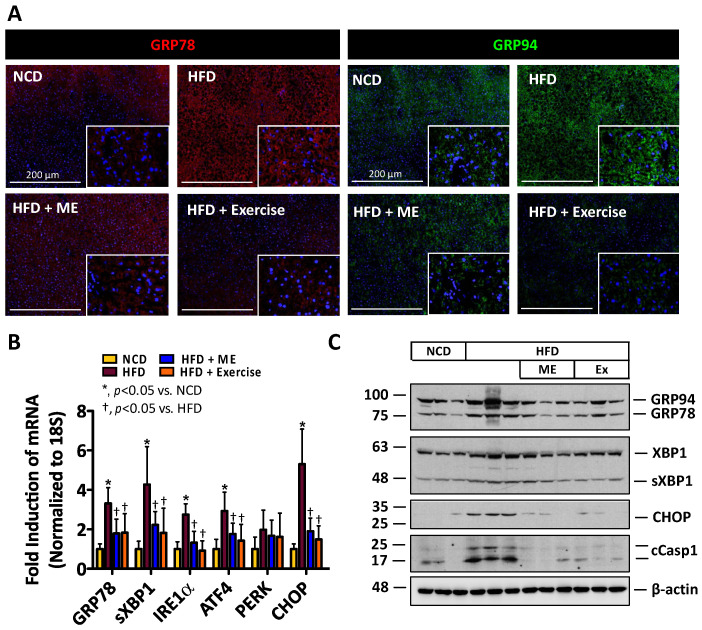
Metabolic enhancer and exercise independently attenuate diet-induced ER stress (**A**) Immunofluorescent staining of GRP78 and GRP94 in the livers of mice fed NCD or HFD supplemented with either ME or exercise. (**B**) Quantitative real-time PCR analysis of hepatic mRNA abundance of indicated ER stress response genes (*n* = 5–6). (**C**) Immunoblot of hepatic protein abundance of indicated proteins (*n* = 3). Scale bars, 200 µm. All data are shown as mean ± SD. *, denotes *p* < 0.05 by one-way ANOVA with Tukey multiple comparison testing using Prism 6 (GraphPad).

**Figure 4 nutrients-15-02410-f004:**
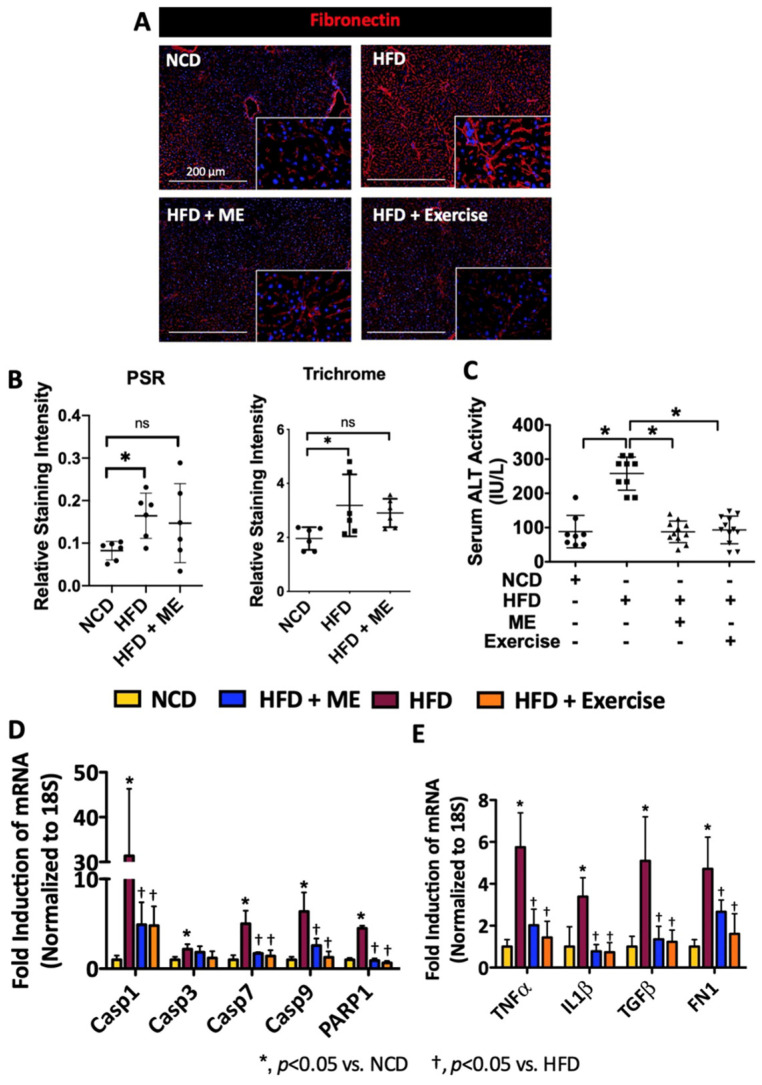
Metabolic enhancer and exercise independently protect against diet-induced liver injury. (**A**) Immunofluorescent staining of fibronectin in the livers of mice fed NCD or HFD supplemented with either ME or exercise. (**B**) Quantification of picro-sirius red (PSR) and trichrome staining livers. (**C**) Analysis of serum ALT activity (*n* = 8–11). (**D**,**E**) Quantitative real-time PCR analysis of mRNA hepatic abundance of indicated genes (*n* = 5–6). Scale bars, 200 µm. All data are shown as mean ± SD. *, denotes *p* < 0.05 by one-way ANOVA with Tukey multiple comparison testing using Prism 6 (GraphPad).

**Figure 5 nutrients-15-02410-f005:**
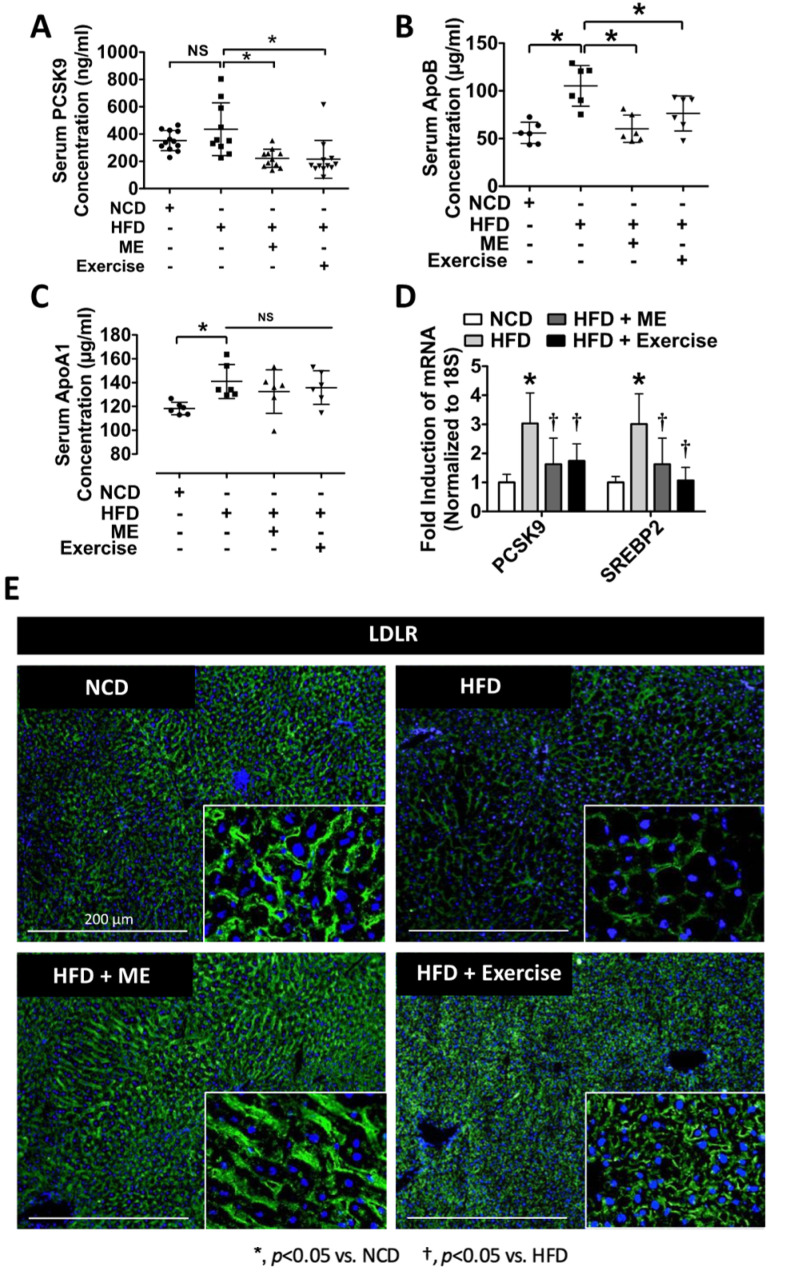
Metabolic enhancer and exercise independently attenuate diet-induced pro-atherogenic serum profile. (**A**–**C**) Serum content of circulating PCSK9, ApoA1, and ApoB (*n* = 8–11). (**D**) Quantitative real-time PCR analysis of hepatic mRNA abundance of indicated genes (*n* = 5–6). (**E**) Immunofluorescent staining of LDLR in the livers of mice fed NCD or HFD supplemented with either ME or exercise. Scale bars, 200 µm. All data are shown as mean ± SD. *, denotes *p* < 0.05 by one-way ANOVA with Tukey multiple comparison testing using Prism 6 (GraphPad).

**Figure 6 nutrients-15-02410-f006:**
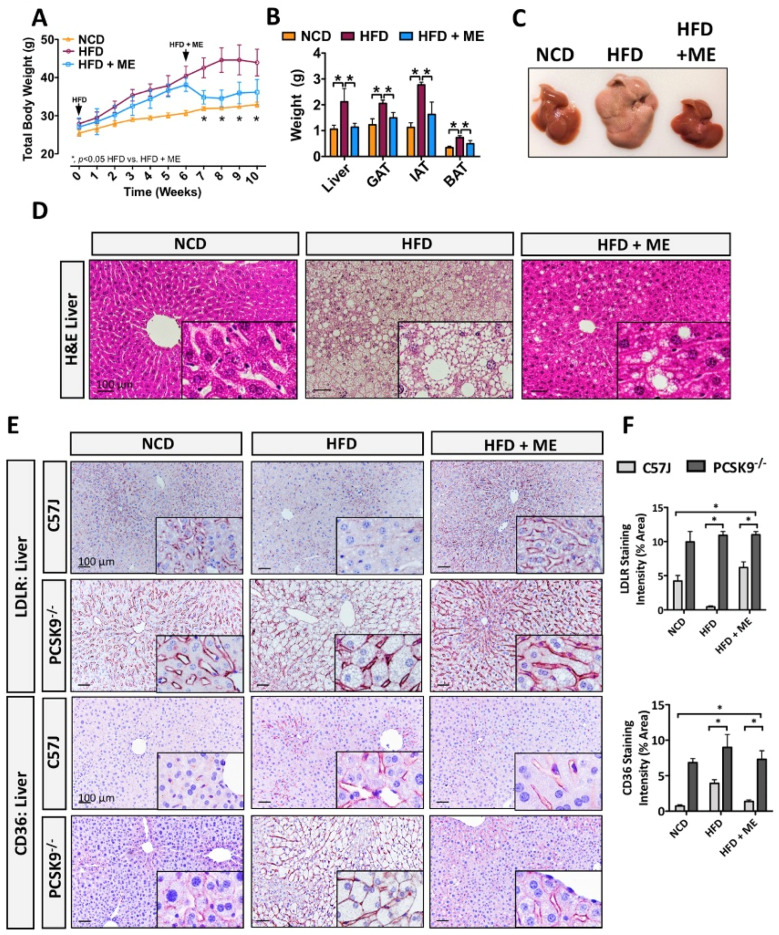
Metabolic enhancer increases hepatic LDLR in a PCSK9-dependent manner. (**A**) Mean body weight and (**B**) mean weight of liver, IAT, GAT, and BAT (*n* = 5). (**C**) Macroscopic appearance of livers from Pcsk9^−/−^ mice fed either NCD, HFD, or HFD+ME (*n* = 5). (**D**) H&E staining of livers. (**E**) Immunohistochemical staining of hepatic LDLR and CD36. Scale bars, 100 µm. All data are shown as mean ± SD. *, denotes *p* < 0.05 by one-way ANOVA with Tukey multiple comparison testing using Prism 6 (GraphPad).

## Data Availability

Not applicable.

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
