# Peer review of "A Metabolic Enhancer Protects against Diet-Induced Obesity and Liver Steatosis and Corrects a Pro-Atherogenic Serum Profile in Mice"

_nutrients, 2023, doi:10.3390/nu15102410_

Round 1

Reviewer 1 Report

Dear authors:

Manuscript entitled "A metabolic enhancer protects against diet-induced obesity, liver steatosis and corrects a pro-atherogenic serum profile in mice" is well-written and reports significant and intriguing data regarding the comparing effects of a metabolic enhancer and exercise on liver steatosis. However, there are suggestions to be addressed:

1_ Line 84: Before introducing PCSK9, you discussed previous findings showing PCSK9 results in hepatic steatosis. For a better understanding, please first explain what is PCSK9 (eg., an enzyme expressed in the liver and ...) then perhaps lines 88-91 should go right after the sentence explaining PCSK9.

2_ Line 134: How ME was given to the animals? Were the ingredients given separately or as individual components (capsule or pill)?

3_ Line 338: wasn't attenuated?

4_ Line 349: You can not conclude that you showed any synergistic effect of ME before measuring every single compound used in ME and running statistics. 

5_ Line 395: ROS were not measured here.

Author Response

1_ Line 84: Before introducing PCSK9, you discussed previous findings showing PCSK9 results in hepatic steatosis. For a better understanding, please first explain what is PCSK9 (eg., an enzyme expressed in the liver and ...) then perhaps lines 88-91 should go right after the sentence explaining PCSK9.

We agree with the reviewer and have restructured the paragraph. PCSK9 is now first introduced for it’s role in the liver as a regulator of several genes; then it’s implications in fatty liver are explained.

2_ Line 134: How ME was given to the animals? Were the ingredients given separately or as individual components (capsule or pill)?

Lines 137 to 145 provide an in-depth explanation of the procedure surrounding the diet. The ME was included in the diet (in the feed) and it’s components are also listed. This is worded as “mice were placed on custom-formulation HFD containing 7 metabolic regulatory and mitochondrial enhancing agents”

note - no change was made to the text in response to this comment

3_ Line 338: wasn't attenuated?

Changed wording to “significantly reduced” to increase clarity.

4_ Line 349: You can not conclude that you showed any synergistic effect of ME before measuring every single compound used in ME and running statistics. 

We agree with the reviewer - the word “synergistic” has been replaced with more suitable terminology throughout.

5_ Line 395: ROS were not measured here.

The reviewer is correct. We have updated the sentence to reflect this remark.

Reviewer 2 Report

The article is well structured and clear.

Important analyzes were carried out.

I request that the authors better describe the characteristics of the diet used, showing the chemical composition and the amount of calories consumed;

It is also important to make clear in the article the total experiment time.

I ask the authors to elaborate a graphical abstract of the manuscript.

References must be adjusted. See the template available on the page in the journal.

The article is well structured and clear.

Important analyzes were carried out.

I request that the authors better describe the characteristics of the diet used, showing the chemical composition and the amount of calories consumed;

It is also important to make clear in the article the total experiment time.

I ask the authors to elaborate a graphical abstract of the manuscript.

References must be adjusted. See the template available on the page in the journal.

Author Response

I request that the authors better describe the characteristics of the diet used, showing the chemical composition and the amount of calories consumed;

The description of the ME diet used in the studies is detailed in lines 138-143 of the methods section. It contained 7 components, in addition to the conventional HFD diet (ENVIGO TD.06414). The chemical composition of the diet, with a breakdown of the proportion of each additional component is also included.

It is also important to make clear in the article the total experiment time.

The experimental timepoints for each mouse experiment are outlined in the methods section (lines 136-139).

I ask the authors to elaborate a graphical abstract of the manuscript.

We agree with the reviewer that a graphical abstract will add much value. A graphical abstract has been added in response.

References must be adjusted. See the template available on the page in the journal.

Formatting for the references has been updated.

Round 2

Reviewer 2 Report

The article can be accepted for publication.

I did not find the requested graphical abstract.

The article can be accepted for publication.

I did not find the requested graphical abstract.

Author Response

Please see attached for the graphical abstract. Note that the following text must be added to section 2.10 Statistical analysis: 

"The graphical abstract was created with BioRender.com (agreement number TS25DLOLAJ)."